# The Impact of Probiotic *Bifidobacterium* on Liver Diseases and the Microbiota

**DOI:** 10.3390/life14020239

**Published:** 2024-02-08

**Authors:** Gabriel Henrique Hizo, Pabulo Henrique Rampelotto

**Affiliations:** 1Graduate Program in Gastroenterology and Hepatology Sciences, Universidade Federal do Rio Grande do Sul, Porto Alegre 90035-003, Brazil; 2Bioinformatics and Biostatistics Core Facility, Instituto de Ciências Básicas da Saúde, Universidade Federal do Rio Grande do Sul, Porto Alegre 91501-907, Brazil

**Keywords:** probiotics, microbiome, microbiota, microbiology, non-alcoholic fatty liver disease, alcoholic liver disease, hepatocellular carcinoma

## Abstract

Recent studies have shown the promising potential of probiotics, especially the bacterial genus *Bifidobacterium*, in the treatment of liver diseases. In this work, a systematic review was conducted, with a focus on studies that employed advanced Next Generation Sequencing (NGS) technologies to explore the potential of *Bifidobacterium* as a probiotic for treating liver pathologies such as Non-Alcoholic Fatty Liver Disease (NAFLD), Non-Alcoholic Steatohepatitis (NASH), Alcoholic Liver Disease (ALD), Cirrhosis, and Hepatocelullar Carcinoma (HCC) and its impact on the microbiota. Our results indicate that *Bifidobacterium* is a safe and effective probiotic for treating liver lesions. It successfully restored balance to the intestinal microbiota and improved biochemical and clinical parameters in NAFLD, ALD, and Cirrhosis. No significant adverse effects were identified. While more research is needed to establish its efficacy in treating NASH and HCC, the evidence suggests that *Bifidobacterium* is a promising probiotic for managing liver lesions.

## 1. Introduction

In recent years, several studies have focused on the relationship between human microbiota and the pathophysiology of liver diseases [1]. The concept of the gut–liver axis is particularly relevant in this regard. It acts as a feedback cycle where changes in the composition of the intestinal microbiota directly affect liver health [2]. The portal vein makes the association between the gut and liver possible, allowing for transporting substances produced by the gut microbiota to the liver. At the same time, the liver secretes bile and antibodies to the intestine [3]. This link is characterized by a high degree of complexity and dynamism. It involves a cascade of biochemical and immunological events related to intestinal barrier permeability, cytokine release, and lipid metabolism molecules [4].

The complex interplay within the gastrointestinal system leads to dysbiosis—an imbalance in the intestinal microbiota [5]. Various studies have suggested that individuals with NAFLD, NASH, ALD, Cirrhosis, and Hepatocellular HCC exhibit dysbiosis [6,7]. Therefore, understanding the microbiota composition and its implications in other biological and clinical aspects can advance the diagnosis and treatment of significant liver lesions [8].

The genus *Bifidobacterium* is a crucial player in the intestinal microbiota. It has several positive effects, such as adhering to the intestinal mucosa, which helps control the permeability of other bacteria, nutrients from food, and molecules resulting from cellular metabolism. Additionally, it produces short-chain fatty acids (SCFA) that play a crucial role in liver metabolism. *Bifidobacterium* also helps control the proliferation of pathogenic bacteria by producing bifidocins and other substances [9]. However, it is known that *Bifidobacterium* levels are reduced in liver diseases, whether alcoholic or nonalcoholic, due to various pathophysiological changes [10,11]. Therefore, the potential of *Bifidobacterium* as a probiotic has been extensively studied. As a live microorganism, it can provide health benefits to the host when administered in the right amounts and at the correct times [12]. The central hypothesis is that *Bifidobacterium* supplementation can reduce dysbiosis and normalize various biochemical and clinical parameters [13].

The most appropriate methods for microbiota studies are next-generation sequencing NGS technologies. The two most common approaches are shotgun metagenomic sequencing and amplicon sequencing. While the shotgun focuses on sequencing entire genomes, the amplicon targets specific genome regions, such as the 16S ribosomal RNA gene (16S rRNA) [14]. Both approaches have supported studying bacterial communities in liver diseases for several years. In this work, we focused on evaluating studies that used NGS technologies to assess the use of *Bifidobacterium* as a probiotic in animal models or human patients with NAFLD, NASH, ALD, Cirrhosis, and HCC. We assessed the main impacts of *Bifidobacterium* use on the composition of the intestinal microbiota as well as biochemical and clinical parameters to understand its beneficial potential and the molecular mechanisms involved.

## 2. Materials and Methods

### 2.1. Search Strategy and Selection Criteria

To conduct the systematic literature search, we established the following cumulative inclusion criteria: only original studies published in the English language from June 2013 to June 2023, with data available in PubMed, Embase, and Web of Science databases, and that used NGS technologies (amplicon or shotgun) in their methodology. We used specific search terms such as “Bifidobacterium”, “NAFLD”, “NASH”, “ALD”, “cirrhosis”, and “HCC”, along with their medical descriptors. We standardized the search strategy for each database to achieve greater efficiency in the search. The full search strategy is available in Appendix A.

The Rayyan tool was expertly employed to identify and remove duplicate articles from multiple databases [15]. After the initial duplicate filtering stage, a meticulous analysis of the titles and abstracts of the remaining articles was conducted to verify and select only the most deserving articles for inclusion. Finally, the studies included in this review were meticulously organized and made available in Appendix A. This review strictly followed the Preferred Reporting Items for Systematic Reviews and Meta-Analyses (PRISMA) criteria [16], as shown in Appendix A.

### 2.2. Data Extraction and Risk of Bias Analysis

We have extracted and organized the data of interest from the studies included in this systematic review in Appendix A. Our analysis includes the primary author’s name, publication year, country of origin, studied liver injury, type of population (human or animal model), type of study (randomized clinical trial or pre-clinical study), sample type, NGS technology, sequencing type, and specific region of the 16S rRNA gene in the case of amplicon sequencing. Additionally, we have included sample size (N), group names, specific taxonomy used as a probiotic intervention, implications on the microbiota, biochemical implications, clinical implications, differentiated-groups approach, and database.

Finally, we conducted a rigorous risk of bias analysis using the Risk of Bias 2 (RoB 2) questionnaires for randomized clinical trials and the Systematic Review Center for Laboratory Animal Experimentation (SYRCLE) for studies with animal models [17,18].

## 3. Results

### 3.1. Selected Studies and Quality Assessment

Initially, 648 articles were identified in the databases. Following the removal of duplicates, 427 articles remained. Through comprehensive analysis, we narrowed it down to 12 articles that met our criteria, as illustrated in Figure 1. Our review was specifically designed to focus on studies utilizing NGS technologies for microbiota analysis, which impacted the sample size. As a result, we only found studies that focused on injuries related to ALD, Cirrhosis, and NAFLD.

We have organized the studies into two tables. Table 1 provides general information about the 12 studies included in the review while Table 2 focuses on the technical aspects of using *Bifidobacterium* as a probiotic (such as its impact on microbiota, biochemical parameters, and clinical parameters).

Table 1 presents details related to the author’s name, publication year, country of origin, population, sample type, sample size, group distributions, and the type of liver injury assessed. Most of the studies were conducted in China, a country with significant academic production in microbiota in liver diseases. Additionally, most studies used animal models for experimental intervention and manipulation. Since the study focused on using *Bifidobacterium* as a probiotic intervention, several species of *Bifidobacterium*, such as *B. breve*, *B. lactis*, and *B. bifidum*, were observed. Some studies used amplicon sequencing, while two studies performed shotgun sequencing. The V3–V4 region was the most commonly used, followed by V4–V5. All studies met the recommended quality criteria, including control or placebo groups for effective comparison alongside intervention groups.

Table 2 summarizes the effects of *Bifidobacterium* intervention on microbiota, biochemical, and clinical parameters. We found that specific bacterial genera experienced a decrease in levels while others exhibited an increase. Most studies used the Linear Discriminant Analysis Effect Size (LEfSe) method to assess differences in microbiota composition between groups. However, some studies used more straightforward methods like the Kruskal–Wallis and Mann–Whitney tests. Although significant variation was in the databases used for taxonomic determination, GreenGenes, SILVA, and RDP were the most common. It is concerning that four studies did not report the database used, and two did not specify the differentiated-groups approach.

Finally, Figure 2 summarizes the data on sequencing type, database, method of abundance differentiation between groups, NGS technology used, and the respective number of times they were identified.

### 3.2. Risk-of-Bias and Quality Assessment

Figure 3 presents the risk-of-bias analysis for the 12 selected studies. Four randomized clinical trials were included in the study and evaluated using the RoB2 tool (Figure 3A). The SYRCLE tool evaluated the eight animal model studies (Figure 3B). Clinical studies show a low risk of bias. In contrast, animal model studies exhibit a high risk of bias.

## 4. Discussion

To provide a comprehensive understanding of the topic, the discussion was organized into the following sections: *Bifidobacterium* as a probiotic (Section 4.1); *Bifidobacterium* in ALD (Section 4.2); *Bifidobacterium* in NAFLD (Section 4.3); *Bifidobacterium* in NASH (Section 4.4); *Bifidobacterium* in Cirrhosis (Section 4.5); *Bifidobacterium* in HCC (Section 4.6); and Quality Assessment and Risk of Bias (Section 4.7).

Collectively, the articles reviewed in this study all aligned in the same direction: *Bifidobacterium* acts beneficially as a probiotic in NAFLD, ALD, and Cirrhosis. It does so by restoring the intestinal microbiota balance, reducing inflammation and oxidative stress, and improving the clinical profile of the gastrointestinal system. However, we did not find studies using NGS technologies to evaluate the effects of *Bifidobacterium* on the pathophysiology of NASH and HCC lesions. Finally, we also discussed the technical aspects of NGS methodologies, bioinformatics, and databases used in microbiota analysis.

Since the search strategy used in this study was comprehensive and well-structured, the absence of studies using NGS technologies to evaluate the effects of *Bifidobacterium* on the pathophysiology of NASH and HCC lesions is indeed a genuine gap in the literature. Future research could explore potential directions for using 16S analysis in clinical and preclinical studies to address this gap. This approach could provide valuable insights into the effects of Bifidobacterium on the pathophysiology of NASH and HCC lesions, filling the current research gap and contributing to a more comprehensive understanding of the role of probiotics in these conditions. Furthermore, future studies could aim to uncover genomic, transcriptomic, and epigenomic signatures associated with these liver conditions, providing a deeper understanding of their pathophysiology. In addition, using targeted metagenomics through 16S analysis is a cost-effective approach for an initial exploratory investigation. Finally, collaborative efforts integrating NGS data with clinical information may pave the way for precision medicine approaches in diagnosing and treating NASH and HCC.

### 4.1. Bifidobacterium as a Probiotic

The term “probiotic” has Greek origins and means “for life”. It was first introduced by Lilley and Stillwell in 1965 and has since expanded to refer to the live microorganisms that help balance the intestinal microbiota [31]. Although probiotics are not exclusively bacteria, they are the primary players, particularly the *Bifidobacterium* and *Lactobacillus* genus [32]. Natural probiotic foods like yogurt, milk, and cheese are rich in these beneficial bacteria [33]. However, with advanced technology, dietary supplements and fecal microbiota transplants (FMT) have emerged as effective methods to improve intestinal microbiota and enhance people’s quality of life [34].

*Bifidobacterium* is an essential genus of bacteria predominantly anaerobic and naturally present in the human and animal intestinal tract. With over 90 species of bacteria, *Bifidobacterium* is ideally suited for the intestinal environment [35]. To be classified as a probiotic, a microorganism must have the ability to colonize a specific location in the host and interact effectively with the host and the host’s microbiota. *Bifidobacterium* fulfills these criteria and is an incredibly valuable probiotic that can enhance the health and well-being of its host [35].

*Bifidobacterium* is a highly beneficial bacteria with many health benefits, including anti-infectious, anti-inflammatory, anti-tumoral, and fat-reducing activities [36]. Its first line of defense is due to its remarkable ability to prevent pathogenic bacteria such as *Escherichia coli* and *Salmonella typhimurium* from lodging in the intestinal mucosa [37]. Moreover, *Bifidobacterium* can significantly increase the levels of *Faecalibacterium*, which is widely known for promoting respiratory tract benefits [38]. *B. pseudocatenulatum*, a specific type of *Bifidobacterium*, has been proven to reduce nitric oxide release, significantly improving vascular function [39]. *B. animalis*, on the other hand, can boost the activity of leukocytes and macrophages while increasing levels of Immunoglobulin A (IgA), which is highly beneficial for protecting the intestinal mucosa [13]. Finally, *B. breve* and *B. longum* are known to stimulate T cells and natural killer cells, both vital immune system components [40].

Probiotics exhibit anti-tumoral activity by eliminating chemical agents that induce carcinoma, altering pH conditions through metabolic activities, and releasing protective anti-mutagenic molecules from the immune system [41]. Studies have shown that *B. longum* effectively suppresses hepatic and mammary carcinogenesis in animal models, while *B. bifidum* inhibits the growth of mutant cells in colon cancer [42,43].

Probiotics are highly effective in reducing body fat. They accomplish this through various mechanisms, including reducing caloric absorption, altering hormonal levels, changing the expression of proteins associated with lipid metabolism and fat accumulation, and reducing inflammation linked to obesity [44]. *B. lactis* has been proven to reduce fat by reducing glucose intolerance in obese or diabetic animals, while *B. breve* has been observed to alter lipid metabolism and antioxidant response, significantly reducing fat [45,46].

Recently, there have been concerns about the safety of probiotics, particularly those made from lactic acid bacteria such as *Bifidobacterium*, due to the production of toxic substances. However, several countries have studied and commercially produced various probiotic strains following the guidelines of the World Health Organization (WHO), ensuring that the use of *Bifidobacterium* as a probiotic is safe by monitoring parameters such as ammonia levels and erythrocyte hemolysis [47]. Large-scale randomized clinical trials have also shown no adverse events, such as hypoglycemia [48]. In vitro and in vivo analyses have further confirmed the safety of *Bifidobacterium*, as no genetic mutagenicity, platelet aggregation, susceptibility to antibiotics, or mucin degradation have been observed [49].

Therefore, *Bifidobacterium* strains of probiotics are crucial in promoting health, providing substantial benefits. These include defense against pathogens, anti-inflammatory and anti-tumor activities, and fat reduction. Several robust studies have confirmed the safety of these strains and endorsed their therapeutic efficacy. In vitro and in vivo analyses and clinical trials have not shown any significant adverse events, reinforcing the confidence in using *Bifidobacterium*. In summary, these probiotics are emerging as indispensable allies in improving health and promoting overall well-being, underscoring their vital role in balancing the intestinal microbiota.

### 4.2. Bifidobacterium in ALD

Individuals with ALD have been observed to have lower levels of *Bifidobacterium* in their intestinal microbiota compared to the control group. This reduction in *Bifidobacterium* levels is caused by excessive alcohol consumption, which can induce inflammatory processes and other complications [11]. An animal model of ALD found that treatment with human beta defensin-2 (hBD-2), a small antimicrobial peptide that mitigates chronic alcohol exposure, restored *Bifidobacterium* levels by reducing steatosis and hepatocyte death [50]. Similarly, patients with ALD who underwent fecal microbiota transplant (FMT) or were treated with pentoxifylline also had increased levels of *Bifidobacterium* due to improved digestion and immunity [51].

New studies are investigating the use of *Bifidobacterium* as a probiotic treatment for people with ALD. Two of these studies used *B. breve* (CICC6182 and ATCC15700) as an intervention. They found that the families of *Lactobacillaceae* and *Muribaculaceae* were more abundant in the treated group than in the disease group. In addition, the *Firmicutes* phylum was reduced following the probiotic intervention [19,21]. A third study included in this review found that using *B. animalis* ssp. *lactis* (Probio-M8 strain) increased the levels of *A. muciniphila*, *L. reuteri*, and *B. uniformis* while reducing the levels of *Bacteroidetes* such as *B. fragilis* and *P. distasonis* [20]. This result is consistent with the finding that reducing Firmicutes levels and increasing *Bacteroidetes* levels is associated with decreased obesity [52].

All the studies included in the analysis found positive biochemical and clinical outcomes from the administration of *Bifidobacterium*. The primary biochemical changes observed were a decrease in the levels of alanine aminotransferase (ALT) and aspartate aminotransferase (AST) enzymes, a decrease in the cytokines TNF-α and IL-β, and a decrease in total bile acids (TBA) and malondialdehyde (MDA). These changes suggest that *Bifidobacterium* improved the health of the intestinal barrier, reduced inflammation, and increased the antioxidant response [19,20,21].

In summary, *Bifidobacterium* has shown promise as a probiotic for treating ALD. It has restored the gut microbiota, improving fundamental biochemical parameters and overall clinical outcomes.

### 4.3. Bifidobacterium in NAFLD

The term NAFLD has been used for many years, but it is now changing to Metabolic Dysfunction-Associated Steatotic Liver Disease (MASLD). The purpose of this new nomenclature is to replace stigmatized words such as “nonalcoholic” and “fatty” [53]. However, since these changes are recent and gradual, and they occurred after the search period within the scope of this review, we will continue to use the old terminology in the systematic search process.

Several studies have found that people with non-alcoholic fatty liver disease (NAFLD) tend to have lower levels of *Bifidobacterium*. Some researchers have investigated inulin, a soluble dietary fiber in certain foods such as bananas, as a possible treatment for NAFLD. After administration of inulin, *Bifidobacterium* levels increased, leading to improved production of short-chain fatty acids (SCFA) and intestinal barrier integrity [54,55]. In seven studies reviewed, *Bifidobacterium* showed promise as a probiotic for NAFLD treatment by restoring the microbiota balance altered by the disease’s pathophysiology. For instance, the administration of *B. adolescentis* (GDMCC60706, GDMCC60707 and CGMCC14395) led to increased bacterial diversity with a higher abundance of genera such as *Lactobacillus*, *Faecalibaculum*, and *Akkermansia* and a reduction in the *Proteobacteria* phylum [24,25]. Other *Bifidobacterium* species, such as *B. animalis* ssp *lactis* (CCTCC2021050), *B. breve* (CBTBR3), and *B. longum* (BCMC02120), were responsible for increasing the *Firmicutes* phylum, especially *Lactobacillus* species. These data were accompanied by improvements in biochemical parameters such as the antioxidant molecules catalase (CAT), glutathione (GSH), superoxide dismutase (SOD), and malondialdehyde (MDA). Therefore, there was a significant improvement in immune balance, intestinal mucosal health, and hepatic metabolism [26,27,28,29,30].

In summary, the evidence is clear that individuals with NAFLD have lower *Bifidobacterium* levels, leading to a deficient inflammatory and metabolic profile. However, numerous studies have shown that probiotics enriched with *Bifidobacterium* species can effectively restore the intestinal microbiota balance and improve biochemical and clinical parameters. The studies reviewed also reported no adverse effects, which strongly supports the notion that *Bifidobacterium* is not only a safe probiotic but also an effective one.

### 4.4. Bifidobacterium in NASH

It has been observed that chronic inflammation and advanced hepatic fibrosis in NASH lead to reduced levels of *Bifidobacterium* [56,57]. Liver transplant procedures have been proven to be a successful solution in improving *Bifidobacterium* levels and reducing fat [58]. Consuming oligofructose is another effective way to increase *Bifidobacterium* levels in NASH patients. This soluble dietary fiber is abundantly found in various foods, including onions [59]. Recent clinical and behavioral studies have demonstrated that *Bifidobacterium* can serve as a probiotic in treating NASH. Malaguarnera et al. found that patients who received *B. longum* and made positive lifestyle changes improved in critical parameters such as AST, ALT, total cholesterol, HDL, and TNF-α [60]. However, no studies using NGS technologies were found within the scope of this review, even though this technology has gained momentum in recent years for studying the microbiome. Furthermore, there needs to be a clear distinction between liver lesions such as NAFLD and NASH, and they are often treated as synonyms in the literature, which is not accurate.

To summarize, studies have shown that patients with NASH have lower levels of *Bifidobacterium*, which is closely related to the disease’s pathophysiology. One potential solution is to use *Bifidobacterium* as a probiotic to restore these levels. However, after conducting a systematic review, we did not find any studies that evaluated the use of *Bifidobacterium* for treating NASH. Although we believe in the probiotic potential of *Bifidobacterium* in treating NASH based on the available literature, we need further updates from the scientific community to confirm this hypothesis.

### 4.5. Bifidobacterium in Cirrhosis

The literature data shows some variations in the scenario of cirrhosis, which is characterized by a more significant degree of hepatic steatosis. While one study indicates reduced levels of *Bifidobacterium* in cirrhotic patients compared to healthy individuals, two other studies report an increase in *Bifidobacterium* levels [61,62,63]. Using lactitol, belonging to the class of osmotic laxatives, raised *Bifidobacterium* levels in cirrhotic patients by reducing endotoxin biosynthesis and inflammation [64]. In a subsequent study, Lu, H., and collaborators employed *B. longum* as a probiotic in a randomized clinical trial. They observed an elevation in *B. breve* and *C. butyricum* levels and a reduction in *K. pneumoniae*. Consequently, there was an enhancement in lipopolysaccharide (LPS) degradation metabolism, an upsurge in butyric acid levels in peripheral blood, and an ultimate improvement in the inflammatory profile [23]. Parallel findings were reported in an animal model by Gómez-Hurtado. The administration of *B. pseudocatenulatum* (CECT7765) in animals with induced cirrhosis resulted in increased *Firmicutes* levels. Biochemical data indicated a decline in the production of TNF-α and IL-6 cytokines, as well as a reduction in inducible nitric oxide synthase (iNOS), indicative of a comprehensive amelioration in both inflammatory and antioxidant aspects [22].

No adverse effects were found using *Bifidobacterium* as a probiotic in treating cirrhotic individuals. Moreover, this probiotic has proven effective in regulating intestinal microbiota and overall organism improvement.

The observed variations in *Bifidobacterium* levels in cirrhosis may be influenced by multiple factors, suggesting a complex interplay within the gut microbiome. The extent and severity of cirrhosis may impact *Bifidobacterium* levels. Different stages of cirrhosis might exhibit distinct microbial profiles, influencing the abundance of *Bifidobacterium* [65]. Variations in *Bifidobacterium* levels could be associated with demographic factors such as age, gender, and ethnicity. These variables may introduce heterogeneity in the cirrhotic population, affecting the composition of the gut microbiome [66]. Therapeutic interventions and medications administered to cirrhotic patients may influence the gut microbiota, including *Bifidobacterium* levels [67]. Investigating the influence of comorbidities on the gut microbiome could also provide valuable insights. In addition, dietary habits and lifestyle factors, such as alcohol consumption and dietary patterns, can significantly shape the gut microbiota [68]. Analyzing the relationship between *Bifidobacterium* levels and these lifestyle elements is essential.

For future research, it would be valuable to conduct comprehensive studies that take into account these potentially influencing factors. This could involve conducting longitudinal studies with larger sample sizes, different stages of cirrhosis, the use of specific medications, and considering a wide range of demographic and clinical variables to better elucidate the factors contributing to the observed variations.

### 4.6. Bifidobacterium in HCC

A healthy liver can undergo degradation pathways through ALD precursor lesions and hepatic steatosis caused by alcohol or through NAFLD and NASH caused by nonalcoholic factors. This process can progress to a cirrhotic condition and ultimately to HCC. Although this is the most common scenario, it is essential to note that the process is dynamic and can vary from person to person. It is also important to remember that there is a possibility of reversing the condition in many cases [69,70].

Medical research has demonstrated that patients with HCC suffer from a change in their intestinal microbiota, leading to decreased *Bifidobacterium* levels [71,72]. Moreover, a study on non-responders to conventional HCC treatments has shown that Tremelimumab and Durvalumab, immunotherapeutic agents, have produced remarkable results. The study authors suggest that the positive outcomes may be attributed to stabilizing the intestinal microbiota and increasing *Bifidobacterium* levels [73].

No studies meeting our review criteria were found regarding the effectiveness of *Bifidobacterium* as a probiotic in HCC. However, a study conducted outside our search period and utilizing shotgun metagenomic sequencing discovered that administering *B. pseudolongum* improved the microbiota composition and intestinal barrier function [10]. This study also demonstrated that acetate production enhanced biochemical parameters, which subsequently interacted with the short-chain fatty acid receptor 43 (GPR43) via the gut–liver axis. Through this interaction, the inflammatory pathway mediated by IL-6, Janus kinase 1 (JAK1), and the signal transducer and activator of transcription 3 (STAT3), was inhibited, resulting in improved outcomes.

The emergence of liver lesions, specifically HCC, is accompanied by alterations in the microbiota and a reduction in *Bifidobacterium*. Despite a lack of relevant research in our comprehensive search, the previous literature indicates that *Bifidobacterium* shows promising potential as a probiotic.

The absence of studies meeting the review criteria regarding HCC does not imply limited research or a specific exclusion criterion. Instead, it reflects the current state of the available literature, where relevant studies meeting the specified criteria are scarce. For future research, it would be beneficial to explore the potential use of *Bifidobacterium* as a probiotic in the context of HCC. This could involve conducting well-designed clinical trials to investigate the impact of Bifidobacterium on the microbiota composition and intestinal barrier function in individuals with HCC. Additionally, further studies could aim to elucidate the mechanisms through which *Bifidobacterium* may interact with inflammatory pathways, potentially leading to improved outcomes for individuals with HCC.

### 4.7. Quality Assessment and Risk of Bias

After applying SYRCLE, one out of eight animal model studies was found to have some concerns, while seven were determined to have a high risk of bias. Figure 3B illustrates that none of the animal model studies provided information on whether the animal allocation was adequately concealed or if the animals were randomly housed during the experiment, as shown in domains 3 (D3) and 4 (D4). Furthermore, none of the studies provided details on blinding of care, researchers, and assessors, which violates domains 5 (D5) and 7 (D7). Only one study had some concerns in domain 1 (D1) and a high risk of bias in domain 8 (D8) due to its failure to clarify whether the allocation sequence was generated and applied appropriately and whether incomplete outcome data were adequately addressed [22]. In contrast, all randomized clinical trial studies assessed by the RoB2 tools were deemed low risk of bias.

The potential problems and significant biases in studies using animal models could greatly affect the trustworthiness of the findings from such research. Insufficient details about how animals were assigned to groups, their housing arrangements, and whether the researchers were aware of the groupings, raises concerns about the accuracy of these studies. Biases in these areas could result in an exaggerated or underestimated impact of treatments, which could limit the applicability of the findings to human situations. In particular, issues with how the groups were formed and incomplete data about outcomes could distort the overall understanding of the results. To enhance reliability, future research should prioritize transparent reporting of allocation methods, housing conditions, and blinding procedures. Implementing standardized tools like SYRCLE for risk of bias assessment and adopting preregistration of research plans can mitigate biases. Furthermore, promoting the publication of “negative” results and addressing time lag bias can contribute to a more comprehensive understanding of interventions.

It is crucial to understand that the method specifications hold critical insights that determine the reliability of the work. Our primary focus was to analyze various NGS technologies, sequencing methods, databases, and differentiation techniques between groups. These factors are essential to ensure the result’s accuracy and validity.

In this current review, most studies have utilized different regions of the 16S rRNA gene, including V3–V4, V3, V4–V5, and V5–V6. This variability in regions makes it difficult to compare the studies. The 16S rRNA gene, consisting of 1500 base pairs, is widely utilized for microbiota research [74]. Its presence in most bacteria, along with its conserved and variable regions, makes this gene a valuable tool for bacterial taxonomy and phylogeny studies. This gene serves as a universal primer due to these characteristics [75]. In microbiome research, two hypervariable regions are typically analyzed for more precise results. Although the V3–V4 regions are commonly used, the recent literature updates suggest that the V1–V2 regions offer improved sensitivity and specificity [76].

SILVA, Greengenes, and Ribosomal Data Project (RDP) were the databases most frequently utilized in the research. Additionally, four studies did not report the database used [19,23,24,25]. The lack of information in the methodology about the database used and the use of obsolete databases, such as Greengenes, is quite concerning. Most studies failed to specify the version, potentially hindering future research groups from reproducing the findings. These issues have been highlighted in the Appendix A.

Due to the vast complexity of the data, the analysis of the microbiome necessitates more robust and appropriate statistical approaches. The first methodologies researchers used to identify biomarkers were the Kruskal–Wallis and Mann–Whitney tests to detect significant differences in the microbiota among sample groups [77]. Subsequently, they employed Linear Discriminant Analysis (LDA) for additional microbiome analysis, leveraging its mathematical capability to maximize the separation between distinct groups [78]. LEfSe, a particular extension of LDA, was also utilized. The LEfSe technique has been widely employed in numerous studies to assess the differences in microbiota composition among various groups. This approach benefits abundance analysis because it considers specific microbiome features critical in distinguishing between groups [79]. More robust and recommended techniques for analyzing microbiomes are also available, such as Analysis of Composition of Microbiomes (ANCOM) and Analysis of Compositions of Microbiomes with Bias Correction (ANCOM-BC). These advanced techniques are advantageous because they incorporate statistical test integration, establish confidence intervals, and increase computational efficiency in their mathematical calculations [80]. However, none of the studies included in this review used ANCOM/ANCOM-BC. Only six studies used LEfSe to compare taxonomic abundance between groups [21,23,24,25,27,29]. Four studies have relied on simpler statistical methods, such as Kruskal–Wallis and Mann–Whitney [20,22,28,30]. Two have not disclosed the methodology employed for group differentiation [19,26].

All reviewed studies reported the use of NGS technology, most of which utilized well-established MiSeq and HiSeq technologies. In addition to the ability to process multiple samples simultaneously, increasing efficiency and reducing operational costs in microbiome studies, these systems are also known for their accuracy and low error rates, which are crucial for ensuring the reliability of data obtained in 16S microbiome studies. [81,82].

In summary, to achieve excellent study results, it is essential to attenuate any present bias. This can be achieved by selecting an appropriate design and following validated checklists such as RoB2 and SYRCLE. The Strengthening of The Organization and Reporting of Microbiome Studies (STORMS) is an innovative tool crucial for improving the quality of microbiome research and should be used in future works [83]. In addition, research groups should provide detailed disclosures of their methodologies, use databases with updated taxonomy, and employ robust statistical and bioinformatics methods.

## 5. Conclusions

Through an extensive analysis of the existing literature, the study explored the potential of using *Bifidobacterium* as a probiotic to treat liver injuries. The results showed that *Bifidobacterium* was remarkably effective in addressing the imbalance in the gut microbiota of both humans and animals, resulting in improved clinical and biochemical parameters. Furthermore, we evaluated the statistical analyses used to detect differential *Bifidobacterium* abundance among the groups. Future studies on liver disease microbiota require updated databases and statistical/bioinformatic methods to fill the observed gaps in the current literature.

## Figures and Tables

**Figure 1 life-14-00239-f001:**
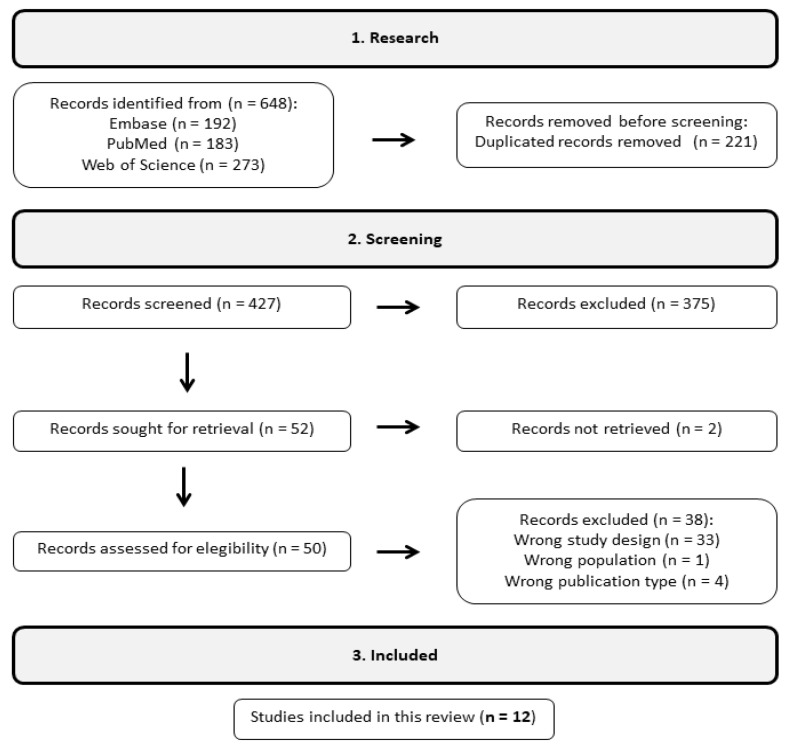
The workflow involved a rigorous filtering process that included 12 studies from an initial pool of 648. This process ensured that the selected studies met the required standards, and that the resulting data could be relied upon confidently.

**Figure 2 life-14-00239-f002:**
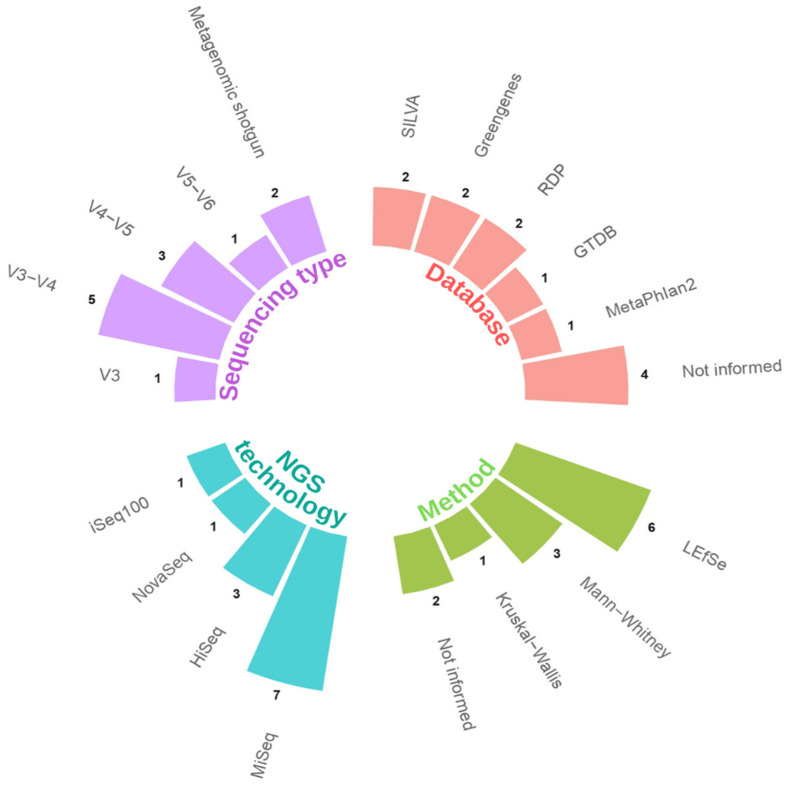
Sequencing type, database, method, NGS technology used, and their respective frequencies. GTDB: Genome Taxonomy Database; LEfSe: Linear Discriminant Analysis Effect Size; RDP: Ribosomal Database Project.

**Figure 3 life-14-00239-f003:**
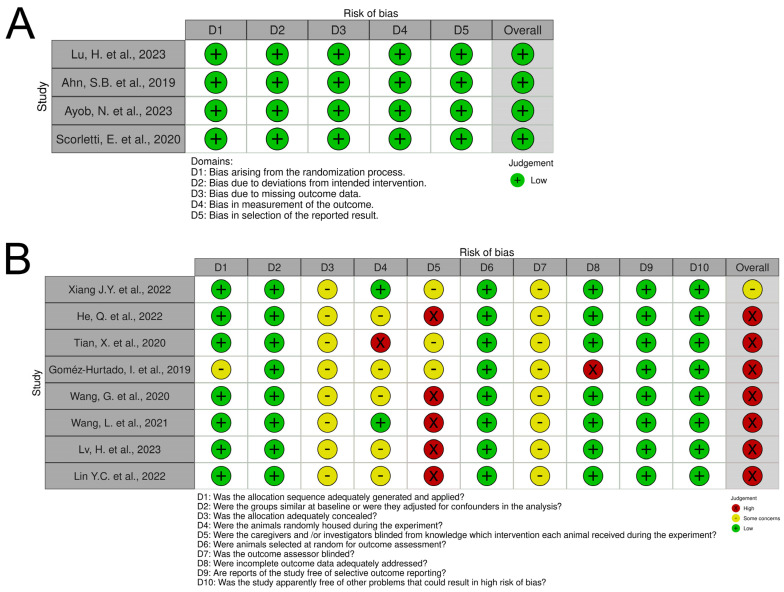
Risk-of-Bias. (**A**) shows the risk of bias in randomized clinical trials [23,28,29,30], whereas (**B**) represents the same for animal studies [19,20,21,22,24,25,26,27]. The letter “D” denotes the domains present in the checklists. Red indicates a high risk of bias, while green signifies a low risk. The yellow indicates “Some concerns”, which serves as a cautionary alert in case the information provided is not clear or was not provided at all.

**Table 1 life-14-00239-t001:** The primary data from the included studies.

Study	Country	Disease	Population	Type of Study	Type of Sample	N	Groups	Probiotic
Xiang, J.Y. et al., 2022 [19]	China	ALD	Animal	Preclinical study	Cecal contents	340	56 Control	*B. breve*
							56 ALD	
							192 ALD + Treatment	
							12 Control + Treatment	
							24 ALD + Probiotic	
He, Q. et al., 2022 [20]	China	ALD	Animal	Preclinical study	Stool	36	9 Control	*B. lactis*
							9 ALD	
							9 ALD + Treatment	
							9 ALD + Probiotic	
Tian, X. et al., 2020 [21]	China	ALD	Animal	Preclinical study	Stool	40	10 Control	*B. breve*
							10 ALD	
							10 ALD + Probiotic	
							10 Probiotic	
Gómez-Hurtado, I. et al., 2019 [22]	Spain	Cirrhosis	Animal	Preclinical study	Stool	20	8 Control	*B. pseudocatenulatum*
							6 Cirrhosis	
							6 Cirrhosis + Probiotic	
Lu, H. et al., 2023 [23]	China	Cirrhosis	Human	Randomized clinical trial	Stool	137	30 Control	*B. longum*
							57 Cirrhosis	
							21 Cirrhosis + Placebo	
							29 Cirrhosis + Probiotic	
Wang, G. et al., 2020 [24]	China	NAFLD	Animal	Preclinical study	Stool	80	8 Control	*B. adolescentis*
							8 NAFLD	
							16 NAFLD + Treatment	
							48 NAFLD + Probiotic	
Wang, L. et al., 2021 [25]	China	NAFLD	Animal	Preclinical study	Stool	108	6 Control	*B. adolescentis* and *B.bifidum*
							6 NAFLD	
							12 NAFLD + Treatment	
							84 NAFLD + Probiotic	
Lv, H. et al., 2023 [26]	China	NAFLD	Animal	Preclinical study	Cecal contents	24	8 Control	*B. animalis*
							8 NAFLD	
							8 NAFLD + Probiotic	
Lin, Y.C. et al., 2022 [27]	Taiwan	NAFLD	Animal	Preclinical study	Stool	24	6 Control	*B. animalis*
							6 NAFLD	
							12 NAFLD + Probiotic	
Ahn, S.B. et al., 2019 [28]	Korea	NAFLD	Human	Randomized clinical trial	Stool	68	36 NAFLD + Placebo	*B. lactis* and *B. breve*
							32 NAFLD + Probiotic	
Ayob, N. et al., 2023 [29]	Malaysia	NAFLD	Human	Randomized clinical trial	Duodenal biopsy	40	22 NAFLD + Placebo	*B. bifidum, B. infantis* and *B.longum*
							18 NAFLD + Probiotic	
Scorletti, E. et al., 2020 [30]	United Kingdom	NAFLD	Human	Randomized clinical trial	Stool	104	49 NAFLD + Placebo	*B. animalis*
							55 NAFLD + Probiotic	

ALD: Alcoholic Liver Disease; NAFLD: Nonalcoholic Fatty Liver Disease.

**Table 2 life-14-00239-t002:** Impact of probiotics on intestinal microbiota, biochemical parameters, and clinical outcomes.

Study	Disease	Probiotic		Microbiota Alterations		Biochemical Outcome	Clinical Outcome
Xiang, J.Y. et al., 2022 [19]	ALD	*B. breve*	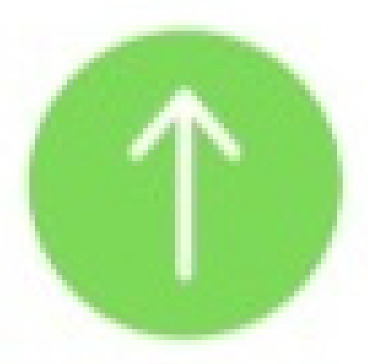	*Lactobacillaceae*	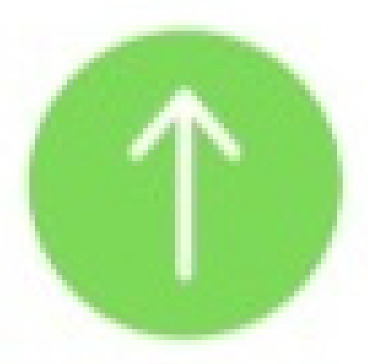	Lactic acid	Enhanced intestinal barrier health attributed to the reduction in pro-inflammatory and oxidative activities
				*Muribaculaceae*			
					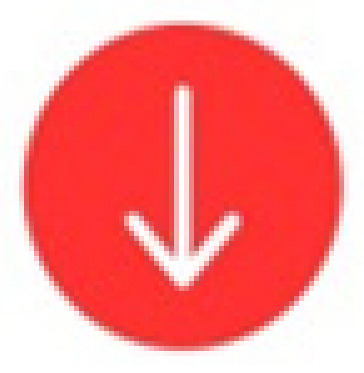	ALT, AST, TNF-α, IL-1β and endotoxin	

He, Q. et al., 2022 [20]	ALD	*B. lactis*		*A. muciniphila*			Improved liver health due to reduced TBA production and levels of pro-inflammatory and oxidative molecules
			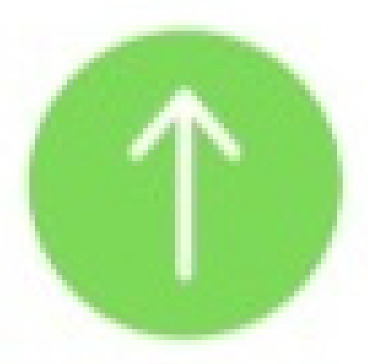	*L. reuteri*			
				*B. uniformis*			
			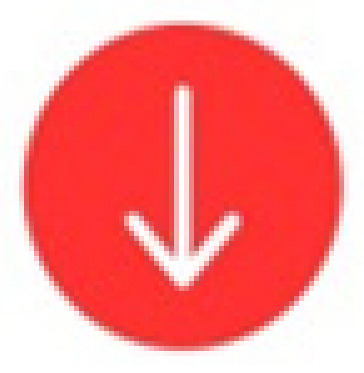	*B. fragilis*	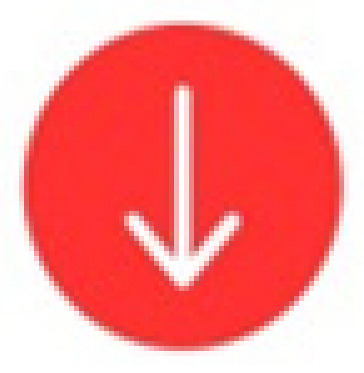	TBA, TNF-1β and MDA	
				*B. thetaiotaomicron*			
				*P. distasonis*			
Tian, X. et al., 2020 [21]	ALD	*B. breve*	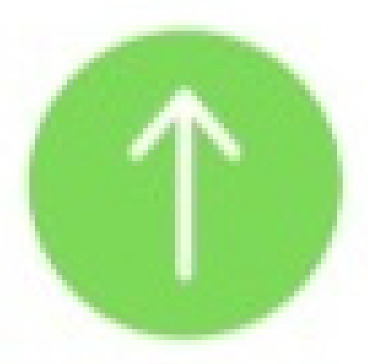	*Bacteroidetes*	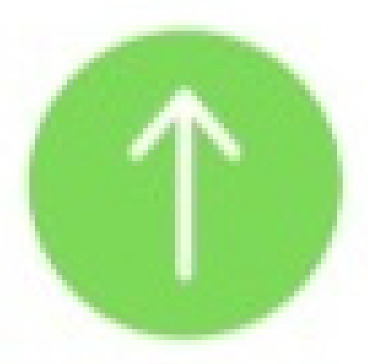	SOD and GSH	Enhanced intestinal barrier, alleviation of endotoxemia, and improved immune homeostasis following alcohol exposure
			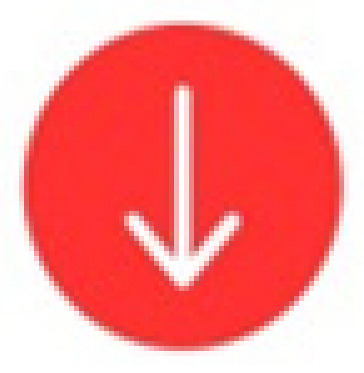	*Firmicutes*	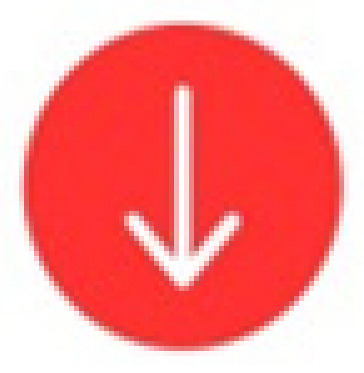	MDA, TNF-α, IL-1β, IL-6 and IL-17	

Goméz-Hurtado, I. et al., 2019 [22]	Cirrhosis	*B. pseudocatenulatum*	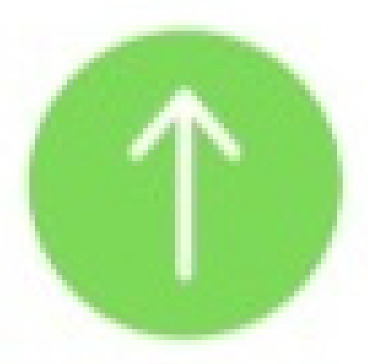	*Bacteroidetes*			Diminished inflammatory processes, leading to a decreased need for antioxidant response
			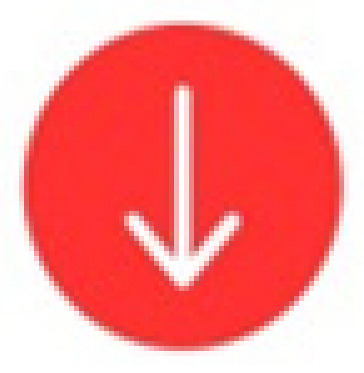	*Proteobacteria*	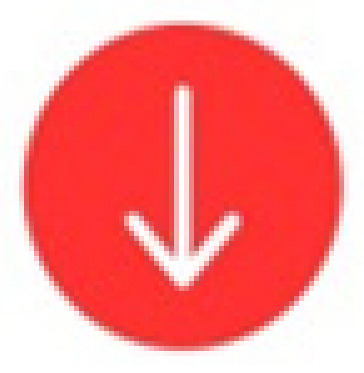	TNF-α, IL-6, iNOS	

Lu, H. et al., 2023 [23]	Cirrhosis	*B. longum*		*B. breve*			Improvement in anti-inflammatory response, immune homeostasis, intestinal dysbiosis, and metabolic defects
				*B. longum*			
			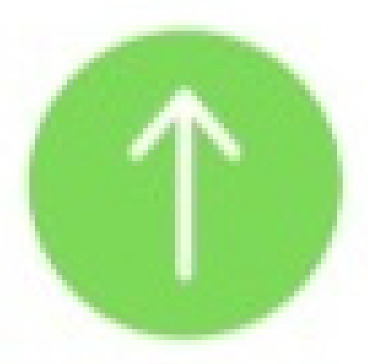	*C. butyricum*	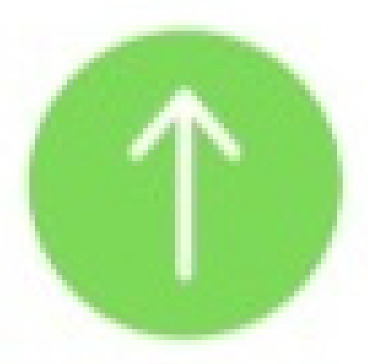	Butyric acid	
				*L. mucosae*			
			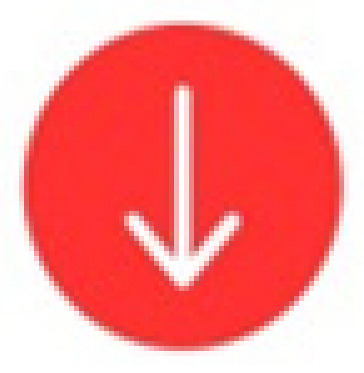	*K. pneumoniae*	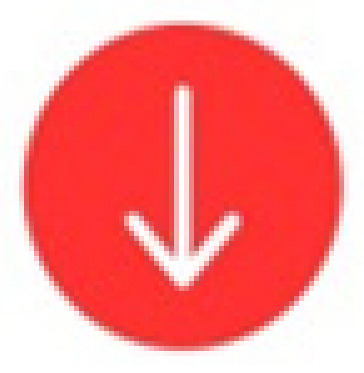	LPS	
Wang, G. et al., 2020 [24]	NAFLD	*B. adolescentis*	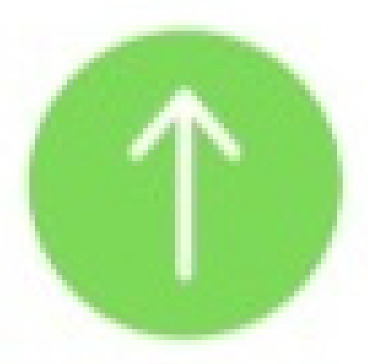	*Bifidobacterium*	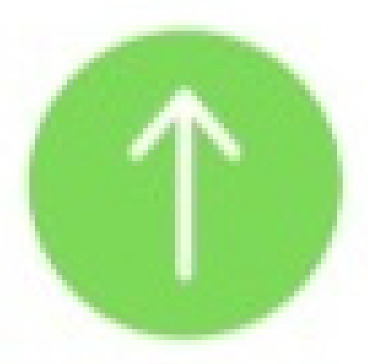	Acetic acid, Propionic acid, Butyric and Isobutyric acids	Improvement in antimicrobial, anti-inflammatory, antioxidant properties, and liver health
				*Akkermansia*			
			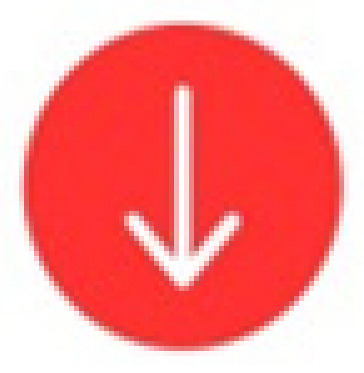	*Clostridium*	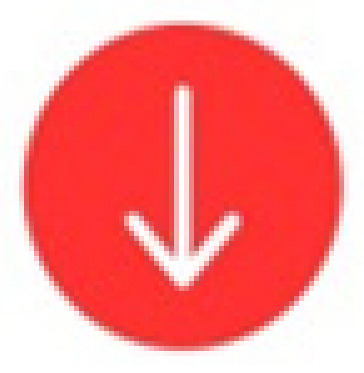	ALT, AST, TNF-α, IL-1β e IL-6	
				*Proteobacteria*			
Wang, L. et al., 2021 [25]	NAFLD	*B. adolescentis* and *B. bifidum*		*Bifidobacterium*			Enhanced inflammatory profile, intestinal permeability, and hepatic metabolism
			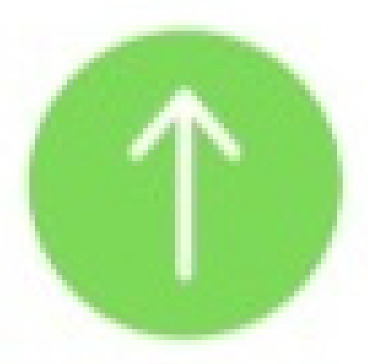	*Lactobacillus*	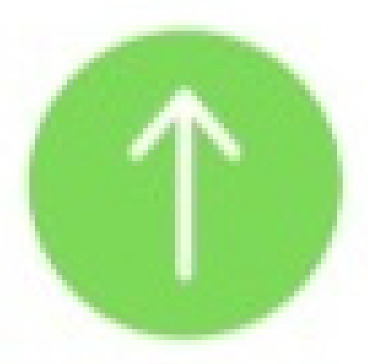	SCFA	
				*Faecalibaculum*			
			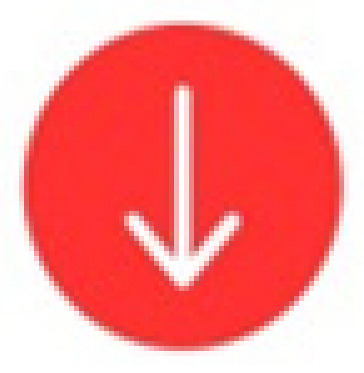	*Ruminiclostridium*	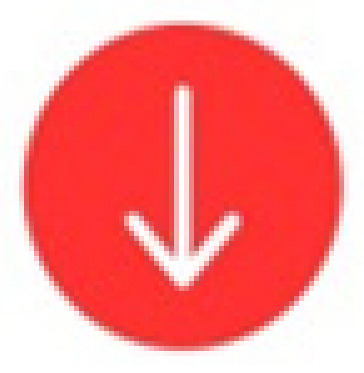	ALT, AST, TNF-α, IL-1β and IL-6	
				*Ileibacterium*			
Lv, H. et al., 2023 [26]	NAFLD	*B. animalis*	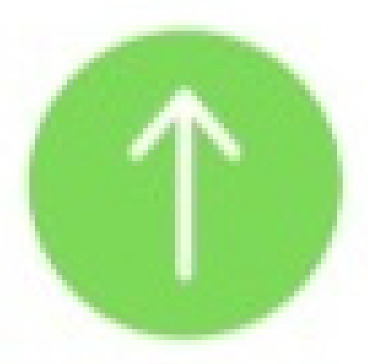	*Firmicutes*	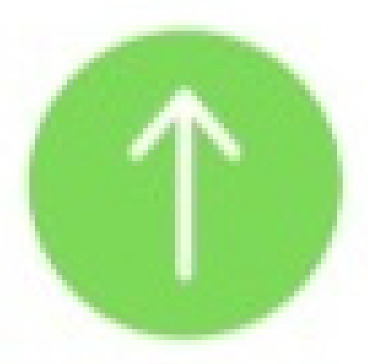	CAT, GSH, SOD and MDA	Improved inflammatory profile, reduced hepatic fat, and enhanced antioxidant response
				*Proteobacteria*			
			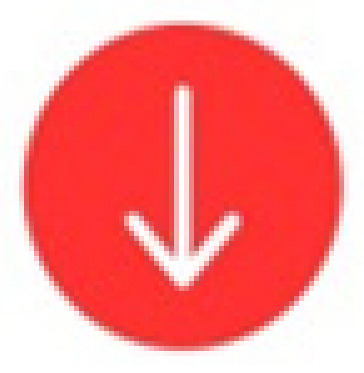	*Fusobacteria*	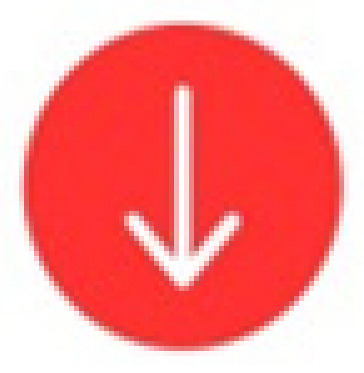	ALT, AST, Hepatocyte adipose area, Insulin, Glucose, NF-κB, TNF-α and IL-6	
				*Akkermansiaceae*			
				*Bacteroidaceae*			
Lin, Y.C. et al., 2022 [27]	NAFLD	*B. animalis*					Improved lipid metabolism and hepatic homeostasis

			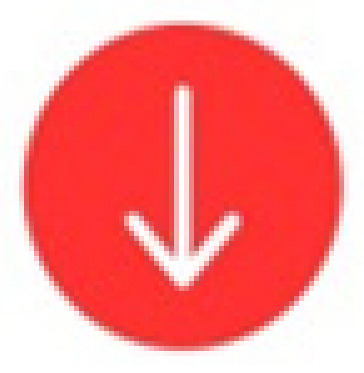	*Desulfovibrionaceae*	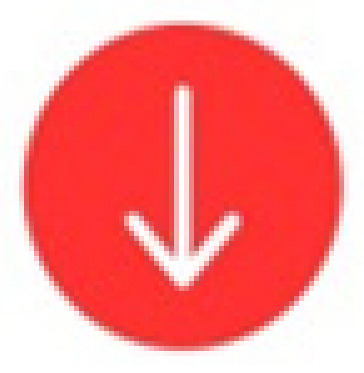	ALT, AST and GGT	
Ahn, S.B. et al., 2019 [28]	NAFLD	*B. lactis* and *B. breve*	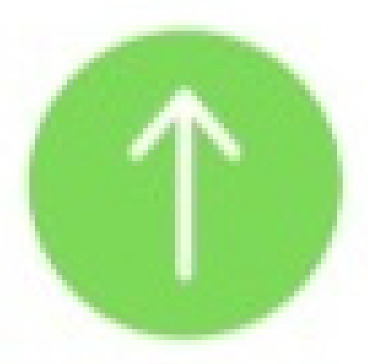	*L. acidophilus*			Improved lipid metabolism and reduction in the pro-inflammatory profile
				*L. rhamnosus*			
					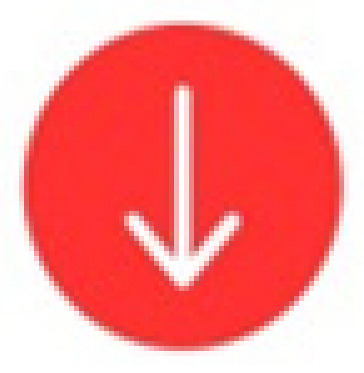	Total cholesterol, Triglycerides and TNF-α	

Ayob, N. et al., 2023 [29]	NAFLD	*B.fidium*, *B. infantis* and *B. longum*	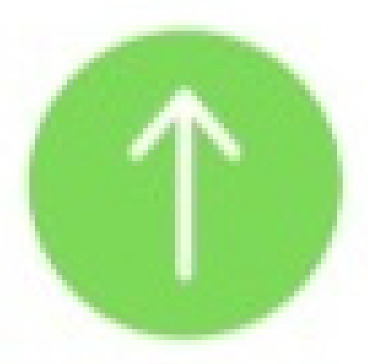	*Actinobacteria*			Enhanced immune homeostasis and intestinal barrier health

					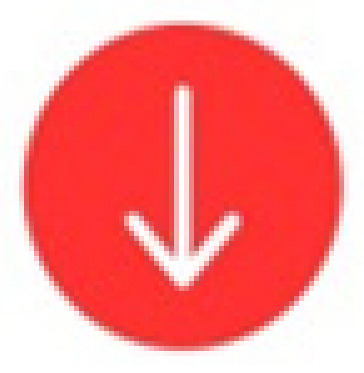	INF-γ, TNF-α and ZO-1	

Scorletti, E. et al., 2020 [30]	NAFLD	*B. animalis*	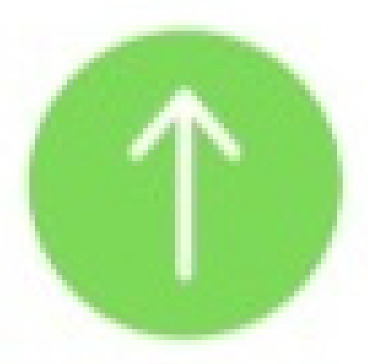	*Bifidobacterium*		No primary outcome was identified	Improved balance of the intestinal microbiota
				*Faecalibacterium*			
			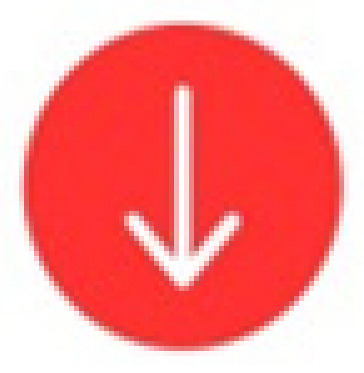	*Oscillibacter*			
				*Alistipes*			

ALT: Alanine Aminotransferase; AST: Aspartate Aminotransferase; CAT: Catalase; GGT: Gamma-Glutamyl Transferase; GSH: Glutathione; IL-17: Interleukin-17; IL-1β: Interleukin-1 Beta; INF-γ: Interferon-Gamma; iNOS: Inducible Nitric Oxide Synthase; LPS: Lipopolysaccharide; MDA: Malondialdehyde; NF-κB: Nuclear Factor-Kappa B; SCFA: Short-Chain Fatty Acids; SOD: Superoxide Dismutase; TBA: Thiobarbituric Acid; TNF-α: Tumor Necrosis Factor-Alpha; ZO-1: Zonula Occludens-1.

## Data Availability

Not applicable.

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
