# Peer review of "The Impact of Probiotic Bifidobacterium on Liver Diseases and the Microbiota"

_life, 2024, doi:10.3390/life14020239_

Round 1

Reviewer 1 Report

Comments and Suggestions for Authors

1. For benefit of student readers, authors may like to elaborate with a line diagram on the anatomy, diversity and importance of 16SrRNA gene in bacteria in the Discussion portion of Page 13.

2. Again for benefit of student readers, authors may elaborate with a line diagram the method used for Linear Discriminant Analysis Effect Size (LEfSe) technique, any precautions of doing it and fallacies if any. 

3. One sentence in material methods is showing Plagiarism "The complete search strategy can be found in Supplementary File 1."

Please change it. 

Author Response

For benefit of student readers, authors may like to elaborate with a line diagram on the anatomy, diversity and importance of 16SrRNA gene in bacteria in the Discussion portion of Page 13.

Reply: We have addressed the necessary updates in lines 458-465 (page 14).

Again for benefit of student readers, authors may elaborate with a line diagram the method used for Linear Discriminant Analysis Effect Size (LEfSe) technique, any precautions of doing it and fallacies if any. 

Reply: We have addressed the necessary updates in lines 475-494 (page 14).

One sentence in material methods is showing Plagiarism "The complete search strategy can be found in Supplementary File 1." Please change it. 

Reply: We have rephrased the mentioned sentence. Please see lines 72-73 (page 2).

Reviewer 2 Report

Comments and Suggestions for Authors

The systematic review titled "The Impact of Probiotic Bifidobacterium on Liver Diseases and the Microbiota" by Gabriel Henrique Hizo and Pabulo Henrique Rampelotto provides a comprehensive analysis of the literature on the potential therapeutic effects of Bifidobacterium in the context of liver diseases. The authors present a well-structured manuscript that follows the Preferred Reporting Items for Systematic Reviews and Meta-Analyses (PRISMA) criteria, ensuring transparency and replicability. The study synthesizes relevant information from a variety of sources and employs advanced next-generation sequencing (NGS) technologies for a thorough investigation.

The introduction is concise; however, it is essential to incorporate recent updates and citations regarding the gut-liver axis to enhance its relevance and reflect the latest advancements in the field.

NGS Technologies in NASH and HCC: The lack of studies using NGS technologies for NASH and HCC is mentioned. However, it's unclear whether this gap is due to a genuine lack of research or limitations in the search strategy. Clarify if the absence of NGS studies is a genuine gap in the literature or if it may be attributed to limitations in the search strategy. Provide insights into potential directions for future research.

Bifidobacterium in ALD: Remark: The section on ALD is well-detailed. However, additional information on the specific strains of Bifidobacterium studied and their mechanisms of action would add depth.

Bifidobacterium in Cirrhosis: Discrepancies in Bifidobacterium levels in cirrhosis are noted. Provide a more detailed discussion on potential factors contributing to these variations. Discuss potential factors causing variations in Bifidobacterium levels in cirrhosis, such as disease severity, patient demographics, or other confounding variables.

Bifidobacterium in HCC: The section on HCC mentions a lack of studies meeting review criteria. Clarify if this gap is due to limited research or a specific exclusion criterion. Clearly state whether the absence of relevant studies is due to a lack of research or specific exclusion criteria. Propose potential avenues for future investigation.

Quality Assessment and Risk of Bias: The quality assessment of animal model studies reveals concerns and high risks of bias. Discuss the potential impact of bias on the overall conclusions. Elaborate on how the identified biases in animal model studies might influence the reliability of the conclusions. Provide recommendations for addressing bias in future research.

The authors are invited to add future directions of the work in the conclusion section.

Overall, the study is well-organized and well-written, yet it is important to address the above-mentioned remarks in your revised version.

Author Response

We thank the reviewer for the useful comments and suggestions. All changes in the manuscript are highlighted in red.

The introduction is concise; however, it is essential to incorporate recent updates and citations regarding the gut-liver axis to enhance its relevance and reflect the latest advancements in the field.

Reply: We have addressed the necessary updates in lines 28-34 (page 1).

NGS Technologies in NASH and HCC: The lack of studies using NGS technologies for NASH and HCC is mentioned. However, it's unclear whether this gap is due to a genuine lack of research or limitations in the search strategy. Clarify if the absence of NGS studies is a genuine gap in the literature or if it may be attributed to limitations in the search strategy. Provide insights into potential directions for future research.

Reply: We have addressed these suggestions in lines 419-428 (page 13).

Bifidobacterium in ALD: Remark: The section on ALD is well-detailed. However, additional information on the specific strains of Bifidobacterium studied and their mechanisms of action would add depth.

Reply: We have addressed these suggestions in lines 284, 288, 317-318, 321 (pages 10-11).

Bifidobacterium in Cirrhosis: Discrepancies in Bifidobacterium levels in cirrhosis are noted. Provide a more detailed discussion on potential factors contributing to these variations. Discuss potential factors causing variations in Bifidobacterium levels in cirrhosis, such as disease severity, patient demographics, or other confounding variables.

Reply: We have addressed these suggestions in lines 375-392 (page 12).

Bifidobacterium in HCC: The section on HCC mentions a lack of studies meeting review criteria. Clarify if this gap is due to limited research or a specific exclusion criterion. Clearly state whether the absence of relevant studies is due to a lack of research or specific exclusion criteria. Propose potential avenues for future investigation.

Reply: We have addressed these suggestions in lines 419-428 (page 13).

Quality Assessment and Risk of Bias: The quality assessment of animal model studies reveals concerns and high risks of bias. Discuss the potential impact of bias on the overall conclusions. Elaborate on how the identified biases in animal model studies might influence the reliability of the conclusions. Provide recommendations for addressing bias in future research.

Reply: We have addressed these suggestions in lines 441-453 (pages 13-14).

Reviewer 3 Report

Comments and Suggestions for Authors

The authors performed a systematic review to investigate how Bifidobacterium could serve as a probiotic in treating different liver conditions such as NAFLD, NASH, ALD, cirrhosis, and HCC, and its influence on the microbiota.

Major issues:

The authors included studies involving both humans and animals. I consider that the studies should be analyzed and presented separately.

Citing articles published (only) in English is scientifically exclusionary. Are data outside Europe and China (and some other Asian countries) not scientifically relevant? This represents a potential serious bias.

I searched for information from (just) Argentina "randomly" in 3 minutes. I don't think the example is relevant to this article - but it is relevant to the mode of selection and this potential bias. ex. doi: 10.1080/19490976.2022.2110821

Minor issues:

Abstract - The authors should explain all abbreviations.

Abstract - The authors should mention that they performed a systematic review.

Introduction - “In recent years, several studies have focused on the relationship between human microbiota and the pathophysiology of liver diseases” – the authors should add references. A little bit longer, with some already well known information

Method - The inclusion and exclusion criteria for the studies presented in this systematic review need to be presented more clearly.

Comments on the Quality of English Language

Please request a native English speaker/writter to revise the manuscript.

Author Response

The authors included studies involving both humans and animals. I consider that the studies should be analyzed and presented separately.

Reply: We rather keep the structure because in each section we followed the same standard of discussing clinical studies first and then the animal studies, which is a smooth way to analyze and present them separately without major section breaks.

Citing articles published (only) in English is scientifically exclusionary. Are data outside Europe and China (and some other Asian countries) not scientifically relevant? This represents a potential serious bias.

I searched for information from (just) Argentina "randomly" in 3 minutes. I don't think the example is relevant to this article - but it is relevant to the mode of selection and this potential bias. ex. doi: 10.1080/19490976.2022.2110821

Reply: We respectfully disagree on this issue. English is the standard language for scientific communication and this criterion represents no bias. In fact, English is an important criterion for systematic reviews because it allows for broader access to scientific literature and helps to minimize bias, ensuring that the review captures a more representative sample of relevant research, ultimately enhancing the robustness and reliability of the review's conclusions. This is in agreement with nearly all high-quality studies (original ones or systematic reviews).

Minor issues:

Abstract - The authors should explain all abbreviations.

Reply: We have corrected the suggested issue in lines 12, 14-15 (page 1).

Abstract - The authors should mention that they performed a systematic review.

Reply: We have corrected the suggested issue in line 11 (page 1).

Introduction - “In recent years, several studies have focused on the relationship between human microbiota and the pathophysiology of liver diseases” – the authors should add references. A little bit longer, with some already well known information

Reply: We have corrected the suggested issue in line 26 (page 1).

Method - The inclusion and exclusion criteria for the studies presented in this systematic review need to be presented more clearly.

Reply: We have corrected the suggested issue in lines 66-69 (page 2).

Round 2

Reviewer 2 Report

Comments and Suggestions for Authors

Great job from the authors for refining the paper's quality, making it a commendable for publication in Life journal. Well done!

Reviewer 3 Report

Comments and Suggestions for Authors

Based on Editors decision, the manuscript is publishable.